# Effect of Warning Labels on Consumer Motivation and Intention to Avoid Consuming Processed Foods

**DOI:** 10.3390/nu14081547

**Published:** 2022-04-08

**Authors:** Cristian Adasme-Berríos, Luís Aliaga-Ortega, Berta Schnettler, Manuel Parada, Yocelin Andaur, Constanza Carreño, Germán Lobos, Roberto Jara-Rojas, Rodrigo Valdes

**Affiliations:** 1Departamento de Economía y Administración, Universidad Católica del Maule, Talca 3460000, Chile; laliaga@ucm.cl; 2Facultad de Ciencias Agropecuarias y Forestales, Scientific and Technological Bioresource Nucleus (BIOREN-UFRO), Centro de Excelencia en Psicología Economía y del Consumo, Núcleo Científico Tecnológico en Ciencias Sociales, Universidad de La Frontera, Temuco 4780000, Chile; berta.schnettler@ufrontera.cl; 3Facultad de Especialidades Empresariales, Universidad Católica de Santiago de Guayaquil, Guayaquil 090150, Ecuador; 4Escuela de Ingeniería Comercial, Universidad Católica del Maule, Talca 3460000, Chile; paradamanuel21@gmail.com (M.P.); yocelinandaur@gmail.com (Y.A.); constanzacarrenoh@gmail.com (C.C.); 5Facultad de Economía y Negocios, Universidad de Talca, Talca 3460000, Chile; globos@utalca.cl; 6Departamento de Economía Agraria, Universidad de Talca, Talca 3460000, Chile; rjara@utalca.cl; 7Escuela de Negocios y Economía, Pontificia Universidad Católica de Valparaíso, Valparaiso 2340000, Chile; rodrigo.valdes@pucv.cl

**Keywords:** nutritional warning labels, front-of-package, processed foods, eating motivation, message, food choice

## Abstract

Nutritional warnings (NWs) as a front-of-package label were implemented as a public policy aiding consumers with recognizing processed foods with high levels of critical nutrients (sodium, saturated fats, carbohydrates, and calories). However, in spite of this tool being well positioned in consumer decision making, there is little extant knowledge about the relationship between the message sent by NW, nutritional knowledge, consumer motivation, and the intention to avoid consuming processed foods. To understand these dimensions’ relations, a theoretical model was created and subsequently tested through structural equations. We applied a survey to 807 home food purchasing decision makers. The results show that the direct effect of NW messages raises the intention to avoid processed foods, while eating motivation is negative in its direct effect on the same avoidance intention. However, the message sent by NWs had a mediating effect between the intentions to avoid processed food and eating motivation but showed no such effect on nutritional knowledge. This suggests that the message sent by NWs was able to turn negative eating motivation into positive eating motivation to avoid processed foods. In conclusion, NWs help mitigate eating motivations, as well as boost the intention to avoid processed foods.

## 1. Introduction

Currently, one of the most important health challenges faced by authorities is obesity. This involves health risks for the population, due to its considerable increase in recent decades [1]. Obesity is associated with the development of non-communicable diseases including diabetes, hypertension, stroke, and certain types of cancer, which impact life expectancy and quality of life (QOL) among the population, along with the associated socioeconomic disadvantages [2]. Problems associated with obesity have led public health authorities to establish mechanisms promoting healthy environments, such as the case of front-of-packaging (FOP) labels on containers. This type of labeling has been used as a tool to inform the population via symbols to stimulate the choice of healthy products [3,4]. Chile used FOP as a regulation to create nutritional warning (NW) labels. This Chilean regulation mandates companies to label foods that surpass established criteria regarding calorie, sodium, sugar, or saturated fat content [5]. Appealing to persuasive communication theory, NWs send a warning message (traffic stop sign), activating risk perception in some people and stimulating the consumption of healthier foods [6,7]. However, the literature is not conclusive in terms of how consumer motivations and their nutritional knowledge interact with NWs. Therefore, the content of the message may generate persuasive effects, motivating changes in behavior and intentions among consumers [8]. Similarly, it has been observed that direct and concrete messages are more effective for motivating the consumption of healthy foods, compared with abstract messages [9]. 

People in charge of promoting nutrition must consider the relationship between motivational factors and information processing, along with the particular area of nutritional knowledge among people [10]. Health-motivated consumers tend to be conscientious about this, and are also involved in nutritional terms, taking proactive measures reflected in buying healthy and high-quality foods, in order to improve their own wellbeing and QOL [11]. On the other hand, among the factors influencing nutritional information comprehension is prior knowledge among individuals [12]. Consumers’ nutritional knowledge can guide attention towards highlighted information in the labeling, along with improving comprehension and influencing food choice [13]. Based on the preceding background, our research aimed to study the existing relation between the message presented by warning labels, nutritional knowledge, consumer motivation, and the intention to avoid consuming processed foods. To achieve this objective, we developed a conceptual model and research hypothesis, which is explained in the following section (see Figure 1).

## 2. Theoretical Framework and Hypothesis

### 2.1. Message Sent by Nutritional Warning Labels

Information theory is concerned with identifying concepts, parameters, and rules which structure message transmission in the context of communication, where information is described via a group of necessary symbols [14]. Among the functions that human communication presents, cognitive or referential function is that which directs the meaning of a message, related to the objective it intends to transmit [15]. In this vein, nutrition and declarations about healthy properties in marketing communications are addressed according to their influence in consumers’ expectations for the food in sensory and non-sensory terms [16]. Public health messages produce behavioral changes in people, contributing to food type choice. However, to achieve this goal, the message must direct action with the goal of increasing the consumption of healthy foods, or at least avoiding the consumption of unhealthy foods [17]. In this sense, from the communicational perspective, labeling delivers a considerable part of the intended message along with legally mandated information.

Labeling information can be defined as an educational message, which seeks to have an effect on the consumer from the perspective of what is socially desirable [18]. NWs send a message, which tries to draw consumers’ attention. Based on principles from persuasive communications theory, Taillie et al. [19] indicate that NWs have the capacity to persuade and are effective when they can gain attention and be precisely understood by people. Thus, NWs do not only fulfill their purpose of providing information about the food product but can also draw consumers’ attention, influencing the decision-making process and subsequent purchasing [20]

Message design can include persuasive calls, indicating desirable behavior for consumers and the benefits. In this sense, both the visual characteristics and the text of NWs can work persuasively by affecting individuals’ behavior, indicating potential health risks and hazards they present [21]. For example, Cabrera et al. [22] found that the color, size, and textual expressions of NWs influence perceived healthfulness and drew the attention of consumers. The information presented on packaging can also shape consumer expectations regarding food elements, such as the case of flavor and the level of health expected [23].

### 2.2. Effect of Nutritional Warning Labels on Consumers

For consumers, nutritional labels are a credible information source, which can be used as a guide for choosing foods in a healthy diet [24]. These labels can influence the decrease in consumption levels of critical nutrients [25]. However, their use can vary by population group, whether the group that may eventually have problems with obesity or overweight status consists of children, adolescents, or adults. Consequently, consumer understanding is key, along with proper nutritional labeling use. Nutritional labels can also work as a warning sign for consumers with other expectations about products, such as their flavor, and for those who do not know about their effects on health [26].

Compared with traditional FOP, NWs can be easier to interpret, promote health, and also discourage consumer purchasing intention [27]. For food purchasing choice makers, NWs (via their design) can correct consumer misconceptions, and also influence on factors associated with beliefs, risk perceptions, and product selection [28]. In this sense, consumers are warned regarding the composition of foods in terms of unhealthy ingredients. However, this information can generate negative attitudes about products with NWs [29].

NWs are an integral strategy for reducing processed food consumption, thereby influencing obesity rate decreases. In this way, we can observe that a simple package with some type of warning label can significantly decrease preferences for sugary drinks in youths and young adults [30]. NWs can better promote health diets, compared with other nutritional schemes applied, due to the consistency between information and communication, and by discouraging consumption of unhealthy foods [31]. 

NWs, therefore, encourage consumers to make healthier food choices, since this type of FOP labeling influences consumer perceptions when foods are hard to recognize from nutritional and health perspectives [32,33].

Based on this background, we propose the following hypothesis:

**Hypothesis** **1** **(H1).**
*The message sent by NWs is positively associated with the intention to avoid processed foods.*


### 2.3. Motivation

Motivation can be defined as an internal process, which regulates and sustains human actions. Its characterization is essential to understanding human behavior [34]. Eating is a vital act for human life but is not unaffected by motivation. In this sense, normal eating behavior is not only related to an absence of dysfunctional feeding processes but also includes motivational elements [35]. This means that eating behaviors include multiple motives, which are evaluated in an everyday way, such as sociocultural factors, social norms, social image concerns, health issues, weight control, and others. In this way, from a psychological perspective, individual motivations can contribute to healthy or disordered eating, arising from positive and negative emotions [36].

Kamrath et al. [11] state that there is a positive relationship between motivation and health decisions since the latter point is connected to a search for active information sources, where consumers with greater health motivation tend to make more use of the information they have available. In this way, consumers with consumption goals regarding nutrient intake and who have health motivations can affect their behavior by paying more attention to food labeling when choosing foods [37]. Messages conveyed by nutritional labels can therefore affect consumers’ eating motivations. One example of this appears in the study by Wegman et al. [23], who found that information written in products’ nutritional information and its indications influence implicit motivation, impacting consumer decision making.

It is also important to add that influences generated in the food selection process are not only tied to health motivations but also include motivational factors from moods, convenience, sensory attraction, natural content, price, weight control, familiarity, and ethical concern [38]. Even in social situations, people are motivated by norms, which affect their food behavior, believing that certain food products are accepted and liked by other people [39]. Motivations for choosing various foods can also vary by the occasion of the day. Motives associated to physiological or functional environments are drivers for food choice in the morning, while psychological or mental motives are associated with moments closer to sunset [40].

From psychology, eating motivation is based on personality traits [41]. So then, the different subdimensions of motivation may arise at the moment of food choice. However, information that captures the consumer’s attention may reduce or impulse the eating motivation to select a type of food. For example, Visschers et al. [42] investigated consumers’ visual attention to nutrition information on food products using an indirect instrument, an eye tracker. This research found that taste motivation overshadows nutrition information; they also found that health motivation may lead to a more profound information process in the food choice. In the same line, Miller et al. [13] found that consumers use food labels; they may be ready to change their diet or incorporate intentions to eat healthier foods. Therefore, following Bialkova et al. [43], the FOP message’s effectiveness depends on health motivation. For example, consumers want to eat cookies, but if they have higher health motivation, they will observe the number of NW before purchasing the product.

Therefore, based on the preceding background the following hypotheses were proposed:

**Hypothesis** **2** **(H2).**
*Consumer motivation affects the intention to avoid processed foods.*


**Hypothesis** **3** **(H3).***Motivation is positively associated with NWs’ message*.

**Hypothesis** **4** **(H4).**
*The message conveyed by NWs has a mediating effect between motivation and the intention to avoid processed foods.*


### 2.4. Nutritional Knowledge 

Nutritional knowledge is a key component of health education, as it is related to food intake and choosing proper diets [44]. In a literature review, Miller and Cassady [13] indicate that nutritional knowledge can be defined as the group of concepts and processes linked to the areas of nutrition and health. Consumers’ individual knowledge relates the health dimension with elements referring to proper diets, food properties, and nutrition-linked diseases. Consumer knowledge is also characterized by the memory domain regarding conceptual and relational-type information [45]. Nutritional education allows people to have information about healthy, balanced diets, which help decrease the risk associated with poor nutrition [46]. In this context, declarations about healthy properties in food products are considered useful by consumers, who value the brevity of messages and their placement on FOP labels. Thus, these declarations about food properties favor both nutritional consciousness and food choice, identifying healthy products [47].

Seeking out nutritional information and knowledge is among the health evaluation predictors for food products [48]. Nutritional labels and NWs, therefore, provide consumers with information, but for purchasing behavior, consumers must be able to interpret labels. In this way, implementing communication messages and practices can promote healthy eating habits. However, low nutritional knowledge among some consumers makes it difficult for them to interpret or comprehend nutritional information contained on labels, which in turn affects food composition evaluation [20]. In this same context, Jáuregui et al. [49] observed that warning labels guide consumers towards healthier options, but participants with low incomes, educational levels, and nutritional knowledge selected less nutritious foods. For their part, Allen and Goddard [50] examined how parents responded to warning labels on cereals with high sugar content, finding that greater nutritional knowledge meant healthier decisions for families. Apart from all of these descriptors relating nutritional knowledge with labeling, another step can be taken since the effect of nutritional labels can be quantified by consumers’ comprehension in direct relation to their nutritional knowledge level [51]. The information available on labels, therefore, facilitates consumers inferring about the health contributions of food products. Miller and Cassady [13], via a cognitive model, related to consumers’ nutritional knowledge and food labeling, identifying the participation of processes related to comprehension including attention, memory, and decision making. These authors indicated that nutritional knowledge allows people to direct their attention towards relevant information, facilitating comprehension, precise knowledge storage, and applying these elements in decision-making situations. For example, a consumer with high nutritional knowledge will be discouraged from consuming food containing two or more NW. Therefore, and in line with the aforementioned background, the following hypotheses are proposed:

**Hypothesis** **5** **(H5).**
*Consumers’ nutritional knowledge affects the intention to eschew processed foods.*


**Hypothesis** **6** **(H6).**
*Nutritional knowledge is positively associated with NWs’ messages.*


**Hypothesis** **7** **(H7).**
*The messages transmitted by NWs have a mediating effect between nutritional knowledge and the intention to eschew processed foods.*


## 3. Methodology

### 3.1. Sample and Procedure

A convenience sample was used, comprised of 807 people who bought processed foods in the city of Talca, Chile, who made the main food choices in their home and met the condition of being legal adults. Data collection also occurred in residential sectors and public places of the city between June and November 2018. The first phase of the survey included a pilot program with ten percent of the sample for questionnaire validation, which allowed us to correct previously undetected problems. Finally, respondents were asked to sign informed consent, previously approved by the Ethics Committee at the Universidad Católica del Maule (Acta N° 85/2017), explaining the goals of the study and that their answers would be treated confidentially.

### 3.2. Questionnaire

The questionnaire had a three-part structure. The first section had general questions about food behaviors, allowing us to identify participants who did not meet the sample filter condition, i.e., excluding individuals who did not consume processed foods and did not make food choices for their households. The respondents selected were the group of individuals who were concerned and conscientious about eating habits’ effects, and the repercussions arising from their choices for the family group as a whole. 

The second part of the instrument sought to evaluate the constructs of nutritional knowledge (NK), message (MSN), and eating motivation (EM) among respondents. The NK dimension included the short scale from Dickson-Spillmann et al. [52], comprised of 20 items evaluating consumers’ nutritional knowledge, even among those who are not entirely familiar with scientific terminology. This line of inquiry considers that greater nutritional knowledge is related to a higher intake of healthy foods, while not being the case for unhealthy foods. The instrument is based on true–false questions, whose sum of correct answers obtained allows for an approximate nutritional knowledge level. Dickson-Spillmann et al. [52] showed a good internal consistency level, with a Cronbach’s Alpha of 0.82, and the average number of correct responses was 13 points within a minimum of 0 and a maximum of 20. These results therefore reflect that consumers have a certain degree of knowledge about nutrition-related topics. The second construct, MSN, is based on constructing a self-created scale via focus groups, considering the perceptions of adult professionals and young students about the message put out by NWs. The scale considers the evaluation of information presented in NWs via 8 items, including: helpful for identifying foods with high amounts of salt, fat, sugar, and calories; helpful for making healthy decisions; helpful for avoiding certain processed foods; helpful for comparing and choosing processed foods; helpful for eating adequate portions; helpful for preventing chronic non-contagious diseases; helpful for avoiding obesity; and helpful for identifying healthy foods more quickly. These items were evaluated on a Likert-type scale ranging from 1 (totally disagree) to 7 (totally agree). The Cronbach’s Alpha for the scale was 0.903. 

For the third construct, EM, we used the instrument developed by Renner et al. [35], which was The Eating Motivation Survey (TEMS) applied to the consumption of processed foods with NWs. TEMS has 45 items and 15 factors allowing for an approximation regarding how consumers choose foods, considering the motivations associated with eating behavior. For each of these items, respondents were asked to give their answers on a seven-point scale (1= never–7 = always). Our study showed a good level of acceptable internal consistency for 12 of the factors (Cronbach’s Alpha > 0.70), while the latent variable of intention to discard (DI), based on the study by Chen [53], corresponds to respondents’ evaluation for discarding processed foods with NWs. This Likert-type scale was represented by four items, such as: I try to avoid eating foods with NWs; I will tell my family members that they should not eat foods with NWs; I will look for information in order to avoid foods with NWs; and I will avoid eating foods with NWs. These four evaluated items go from 1 (totally disagree) to 7 (totally agree), and their reliability level in the Cronbach’s Alpha was 0.849. Finally, for the third section of the questionnaire, we included classification questions that covered information about the demographic factors of gender, age, and educational level.

### 3.3. Statistical Analysis 

To study the effect of the message sent by NWs on the intention to avoid processed foods related to nutritional knowledge and eating motivation of consumers, we initially applied the confirmatory factorial analysis (CFA) technique allowing for factor quantity specification, in order to measure convergent and discriminant validity [54]. In order to identify subdimensions of eating motivation (EM), we also developed a second-order CFA. After verifying the dimensions’ internal consistency, we proceeded to evaluate the hypotheses established in the conceptual model by estimating structural equation models (SEM) from a group, using the Mplus statistical program, version 8.1. This multivariant technique allows us to identify complex relations within conceptual models via road maps and regression-based equation systems. We then proceeded to evaluate the full structural model in two stages, with the specification of the hypotheses representing the coefficients’ paths, derived into a reduced version by ruling out subdimensions from the EM dimension with low factorial loads. 

The model allowed us to identify direct and indirect effects, where nutritional knowledge and eating motivation take on a control condition. In this way, these variables’ effects on the intention to avoid processed foods with NWs are identified in a tertiary form in the conceptual diagram (Figure 1). In this case, the intermediate variable MSN takes on the condition of mediator, showing how these independent variables affect the results of the study. A mediator effect, therefore, arises, representing the step from message to intention to avoid, according to nutritional knowledge and EM in this process. The sample size used in the study (*n* = 807) fulfills the requirements suggested by Koran [55] of 200 cases for a three-factor model.

## 4. Results

### 4.1. Characterization of Variables Based on Sample

The sample was composed of 63.82% women and 36.18% men with a median age of 37 years. A total of 34.07% of the sample finished higher education. Processed food consumption frequency was approximately three products per day. Table 1 shows the median, standard deviation, and correlation of the variables nutritional knowledge (NK), message (MSN), discard intention (DI), and the subdimensions of EM. Regarding the variables studied, results showed that DI for processed foods with NWs had a medium-high score (M = 5.00; SD = 1.68), while NK was medium-high (M = 12.67; SD = 3.33), indicating those study participants generally had some degree of nutritional knowledge about the foods they consumed. In the area of MSN transmitted by NWs, the evaluation was medium-high (M = 5.35; SD = 1.52). These results showed that consumers considered NWs in the food decision-making process.

For the EM subdimensions, we found that the Liking factor (M = 5.04; SD = 1.81) had the highest score within this dimension. This reflects the importance of product composition for processed food consumers, and therefore the relation with their desires and good flavor. In second place, we find convenience (M = 4.60; SD = 1.73), in terms of practical reasons linked to this type of food, such as easy preparation. Pleasure (M= 4.48; SD = 1.82), linked to reward and enjoyment, has a medium-high score. In the area of habits, such as familiarization and the custom of using this type of food, we find a medium score (M= 4.22; SD = 1.91). In the need and hunger motivation, we also find a medium score (M = 4.09; SD = 1.71), regarding the need for energy, satisfaction, and hunger.

The EM subdimensions with the lowest weight included affect regulation (M = 2.48; SD = 1.68), social norms (M = 2.29; SD = 1.50), and social image (M = 1.95; SD = 1.27), implying that at a general level participants do not relate processed food consumption with emotional elements, such as feelings of sadness, frustration, loneliness, or behaviors which disappoint others. The weight control (M = 2.36; SD = 1.47) and health (M = 1.97; SD = 1.37) factors also showed lower weight, representing little concern for health standards, calorie control, and balanced diets. Study participants are therefore influenced by practical and pleasurable motivations for consuming processed foods.

Correlations show strong associations between the EM factors of Need and Habits (r = 0.69 *p* ≤ 0.01), pleasure and liking (r = 0.67 *p* ≤ 0.01), and need and liking (r = 0.65 *p* ≤ 0.01). Regarding DI and EM factors, we observe low and negative relations for effects from liking, habits, need, convenience, and pleasure. However, there are positive medium effects between DI and MSN (r = 0.46 *p* ≤ 0.01), but not with DI and NK (r = 0.07 *p* ≤ 0.01). Therefore, the takeaway is that the greater the effect of the message sent by NWs, the more consumers will tend to avoid consuming processed foods with this label type. 

### 4.2. Model Tests

First, we evaluated the convergent validity of the measuring model (CFA) via factor loadings, and average variance extracted (AVE) (see Table 2). Constructs 3–17 correspond to the first-order CFA of the EM, whose factorial loadings oscillate between 0.55 and 0.93. We subsequently performed a second-order CFA in order to better represent EM. We also calculated the Cronbach’s alpha and compound reliability (CR) which were greater than the suggested theoretical measurement of 0.70. Meanwhile, the AVE of the model was above the recommended level of 0.5, apart from traditional eating which was at 0.44. 

Three model estimates were generated, allowing us to evaluate the factors’ effects on the DI dimension. In this way, all the models had good fit (see Table 3). For the measurement model, the RMSEA had an acceptable fit of 0.044, and the chi-squared degree of freedom relation was 2.561, which is below the theoretical threshold measurement. The comparative fit index (CFI) and Tucker–Lewis Index (TLI), at 0.904 and 0.889, respectively, are next to the theoretical fit adjustment of 0.90.

For the structural model, which includes all the EM subdimensions (as a second-order CFA), we obtained an RMSEA of 0.049, indicating an acceptable fit. The CFI and TLI indices were 0.898 and 0.892, close to 0.90. The chi-squared/df relation had an indicator of 2.669, representing a good fit for the general model in its group. Subsequently, four of the 15 EM factors were ruled out, since the only factors considered were those with factorial loadings greater than 0.4 [54]. These were health, concern about naturalness, weight control, and social image, which can be explained because the study was carried out with a focus on processed food consumption in general and not using specific products. With the eleven subdimensions comprising the EM dimension, the reduced model was evaluated. This new final model had better fit adjustments, slightly decreasing the chi-squared/df ratio to 2.628, and improving goodness of fit indicators for the model.

Table 4 shows the estimate of the direct, indirect, and total effects standardized for reduced model effects. This made it possible to isolate the mediation effect of the message sent by NWs in the proposed model. On the NK side, we found no evidence that this variable influenced the intention to avoid buying foods with NWs by the respondents. Although the total effect is significant, this only corresponds to an additive effect of direct and indirect effects. The EM dimension comprised of various subdimensions also showed evidence of its influence in the removal of processed foods with NWs. 

Upon testing the reduced model, we obtained the effects of the EM and MSN dimensions and the NK variable on the DI dimension, establishing them as a mediator for the effects of KN and EM, for the MSN sent by NWs. In Figure 2, we can see the standardized coefficient paths, where solid lines correspond to direct effects, and the dotted line to indirect effects. 

We found that NWs’ message is positively associated with DI, observed via a significant direct effect (H1; β = 0.535, *p* ≤ 0.001). EM as a second-order dimension whose chosen subdimensions were related with respondents’ eating habits and needs, showed significant results in its direct and indirect effects. In this sense, we found two direct effects. The first is negative with regards to DI (H2: β = −0.099, *p* ≤ 0.01), while the second is positive with regards to MSN (H3: β = 0.114, *p* ≤ 0.01). Finally, we observe that MSN is a dimension which generates a positive mediation effect between EM and DI (H4; β = 0.061, *p* ≤ 0.01). With regards to NK, there was no influence on consumers’ processed food eating decisions, as H5, H6, and H7 could not be proved in this study.

## 5. Discussion

Nutritional warning labels have gained importance worldwide, since their direct message makes them more efficient than other types of FOP labels [56]. However, it is necessary to further study how eating motivations and nutritional knowledge interact with this public policy. In this context, the present study proposed a theoretical model studying the relation of direct and indirect effects between the message delivered by warning labels, nutritional knowledge, consumer eating motivation, and the intention to avoid consuming processed foods. In this section, we will discuss the results obtained by the model in separate sections.

### 5.1. Direct Effects on Intention to Discard Foods with NWs

The message given by warning labels facilitates interpretation of nutritional information and encourages consumers to discriminate between more or less healthy processed foods when taking eating decisions [56,57,58]. In this context, the results of this study show that the message given by NWs directly affects processed food consumers’ eating decisions, with this variable having the greatest weight in the estimated model (H1). This result is consistent with other previous studies, which showed evidence that the black octagon has the most dissuasive effect for avoiding the consumption of foods with high critical nutrient contents, compared with other types of warning labels [19]. This can be because NWs can efficiently draw consumers’ attention, requiring less time to process FOP information schemes, increasing negative affect, and avoiding erroneous perceptions that consumers may have towards some products described as healthy by manufacturers [28,59,60]. 

On the other hand, eating motivation is composed of diverse factors which influence eating decisions [35]. Our findings show that eating motivation is negative in its direct effect on the intention to discard processed foods with NWs (H2). The interpretation of this finding refers to how respondents did not feel the motivation to discard a processed food with NW. This may be because, among the consumers interviewed, the primary motivations mentioned are that processed food is a traditional food for them, that it gives them pleasure, satisfies their needs and hunger, tastes good to them, is fairly priced, allows them to socialize, is convenient for them, is appetizing, and is part of their consumer habits. With less relevance, affective regulations and social norms are part of the eating motivations for these household food decision makers. These findings are in line with previous studies using TEMS, which showed that hedonic, functional, and psychological motivations along with social context are motivational predictors in food choice for different eating occasions [61,62].

Regarding nutritional knowledge, no significant direct effects were present in the intention to discard processed foods by the evaluated consumers. This may be because all interviewees, since they were household food decision makers, felt that they had some amount of knowledge about the components of the processed foods they consumed. In this way, MacArthur et al. [63] found that nutritional knowledge increased with age and education, but not enough for food choice. This can be explained by the superficial use of information by consumers, implying a limited comprehension of the message regarding the nutritional content of the food [64]. A recent study by Koch et al. [65] identified nutritional knowledge areas but concluded that it was unlikely that increasing this knowledge would mean improved dietary behavior.

### 5.2. Mediating Effect of NW Messages on Consumers

Food labels generally send a message generating trust in the end consumer [66]. This trust in the public policy of food labeling is a motivation for considering warning labels in food purchasing [67]. In this context, the results of the present study show that the message given by NWs had a mediating effect between the intention to discard a processed food (DI) and eating motivation (EM), but had no such effect on nutritional knowledge (NK). This suggests that the message sent by NWs has the capacity to turn a negative eating motivation (H2) into a positive eating motivation to discard a processed food (H4). The importance of this finding lies in how the message sent by NWs is the key element influencing household food decision makers (H1), given the persuasive capacity of NWs (both visual and textual) to affect consumers’ eating motivation. This finding fits with the previous study by David et al. [68] who found that NWs influenced decisions to stop the appetizing signals of ultra-processed foods, since this type of FOP warning label generates negative emotions, increasing anticipated social interactions, and keeping in mind the damage that can be generated by the food and changing the social norm, ultimately affecting consumer behavior [69]. However, it must be considered that the effect of NWs over time on choosing processed foods that are less damaging for health will depend on personal motivations and the provision of information to consumers reinforcing healthy food consumption [70]. This public food labeling policy must therefore be constantly reinforced over time to achieve the desired effects favoring decreased obesity.

On the other hand, our findings are highly relevant for public food policy decision makers, since warning labels by themselves are unable to change the eating habits and lifestyles of people. However, constant reinforcement of the message sent by this type of labeling can help consumers to consider warnings not only as a credibility attribute but also as an attribute to search for when choosing food.

The first limitation of this study is that it was based on processed foods generally, and did not consider the motivational effect that one specific processed food can have in consumer food choice, which would open new research options based on the real effect of warning labels on food decisions. Another limitation of this study is that it was conducted via convenience sampling, making it unrepresentative of the general population. However, the sampling was obtained from household food decision makers, reflecting household consumption. These limitations can be addressed in future studies to identify how people manage information about risks associated with food composition and how they use warning labels in the selection process. 

Another limitation is that the study was carried out in Chile, however, the NWs in the food context are being used by countries such as Israel, Uruguay, Perú, Brazil, Colombia, México, Finland, Argentina, and Canada. In spite of the limitations of this study, it contributes a theoretical model which provides evidence about how warning labels can change eating motivation in food selection. In addition, the study helps to understand the role of NWs and other interpretative FOP labeling in food choice to prevent/inform the population about critical nutrients

## 6. Conclusions

NWs influence Chilean consumers’ eating decisions in such a way that the message sent by this labeling increases the intention to discard processed foods. In this sense, the warning represented by notifications about critical nutrient levels presents information affecting the selection process. This reflects the effectiveness of Chilean rules which serve as an orientation for consumers about food properties. On the other hand, we found partial evidence about this effect of NWs regarding the statements about motivation and nutritional knowledge. The message given by the labels affects the motivation dimension, via a mediating effect regarding the intention to discard processed foods. We can take away, therefore, that individuals’ motivations toward processed foods are decreased. On the other hand, we found no evidence for effects on nutritional knowledge, which could be attributed to the fact that all consumers have some amount of general knowledge about food. It is necessary to go deeper into the practices which promote healthy eating habits, considering the role of warning labels and their effects on the different dimensions forming part of the food selection process.

## Figures and Tables

**Figure 1 nutrients-14-01547-f001:**
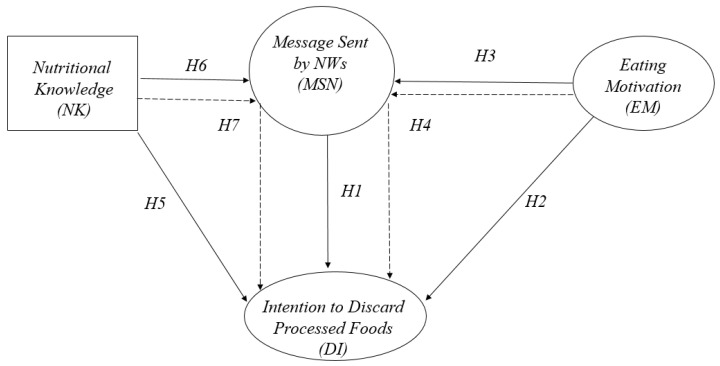
Conceptual Model.

**Figure 2 nutrients-14-01547-f002:**
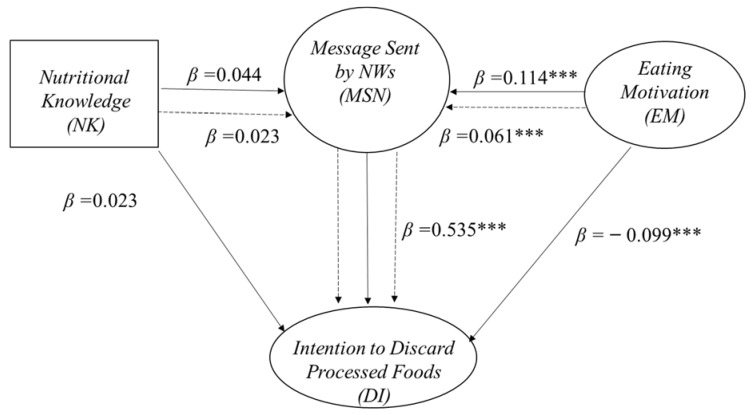
Results of testing the reduced model: standardized path coefficients (β). Note: *** significant at level 0.01.

**Table 1 nutrients-14-01547-t001:** Mean, standard deviation, and Pearson’s correlation between determinants and intention to discard (*n* = 807).

								Correlations												
	Mean	Sd.	CN	MSN	DI	Liking	Habits	Need and Hunger	Health	Convenience	Pleasure	Traditional Eating	Natural Concerns	Sociability	Price	Visual Appeal	Weight Control	Affect Regulations	Social Norms	Social Image
			1.	2.	3.	4.	5.	6.	7.	8.	9.	10.	11.	12.	13.	14.	15.	16.	17.	18.
CN	12.67	3.33	1																	
MSN	5.35	1.52	−0.01	1																
DI	5.00	1.68	0.07 **	0.46 ***	1															
Liking	5.04	1.81	−0.15 ***	0.06	−0.09 ***	1														
Habits	4.22	1.91	−0.27 ***	0.04	−0.13 ***	0.63 ***	1													
Need and hunger	4.09	1.71	−0.25 ***	0.10 ***	−0.11 ***	0.65 ***	0.69 ***	1												
Health	1.97	1.37	−0.17 ***	0.05	−0.06	0.09 **	0.28 ***	0.31 ***	1											
Convenience	4.60	1.73	−0.14 ***	0.03	−0.09 ***	0.51 ***	0.46 ***	0.51 ***	0.12***	1										
Pleasure	4.48	1.83	−0.18 ***	0.08 **	−0.06 *	0.67 ***	0.52 ***	0.62 ***	0.14 ***	0.52 ***	1									
Traditional eating	3.59	1.61	−0.17 ***	0.11 ***	0.06 *	0.46 ***	0.45 ***	0.44 ***	0.22 ***	0.47 ***	0.60 ***	1								
Natural concerns	2.25	1.53	−0.16 ***	0.04	0.00	0.09 ***	0.17 ***	0.24 ***	0.48 ***	0.08 **	0.13 ***	0.22 ***	1							
Sociability	3.74	1.76	−0.24 ***	0.06 *	0.01	0.47 ***	0.37 ***	0.44 ***	0.17 ***	0.41 ***	0.53 ***	0.51 ***	0.31 ***	1						
Price	3.84	1.91	−0.21 ***	0.05	−0.04	0.47 ***	0.51 ***	0.50 ***	0.24 ***	0.50 ***	0.47 ***	0.49 ***	0.24 ***	0.52 ***	1					
Visual appeal	3.06	1.76	−0.14 ***	0.08 **	0.01	0.31 ***	0.35 ***	0.37 ***	0.26 ***	0.33 ***	0.42 ***	0.54 ***	0.29 ***	0.49 ***	0.47 ***	1				
Weight control	2.36	1.47	−0.13 ***	0.10 ***	0.02	0.08 **	0.15 ***	0.19 ***	0.44 ***	0.08 **	0.09 **	0.20 ***	0.54 ***	0.25 ***	0.26 ***	0.34 ***	1			
Affect regulations	2.48	1.68	−0.14 ***	0.12 ***	0.00	0.23 ***	0.24 ***	0.31 ***	0.23 ***	0.15 ***	0.37 ***	0.42 ***	0.21 ***	0.33 ***	0.27 ***	0.41 ***	0.22 ***			
Social norms	2.29	1.5	−0.14 ***	0.09 **	0.06	0.20 ***	0.22 ***	0.20 ***	0.28 ***	0.11 ***	0.26 ***	0.45 ***	0.30 ***	0.37 ***	0.30 ***	0.44 ***	0.38 ***	0.45 ***	1	
Social image	1.95	1.27	−0.11 ***	0.05	0.03	0.01	0.12 ***	0.17 ***	0.34 ***	0.12 ***	0.11 ***	0.31 ***	0.33 ***	0.26 ***	0.26 ***	0.42 ***	0.40 ***	0.39 ***	0.56 ***	1

Note: *,**,*** significant at levels 0.10, 0.05, and 0.01, respectively.

**Table 2 nutrients-14-01547-t002:** Standardized factorial CFA loadings in the first and second order, composite reliability estimates, and average variance extracted (*n* = 807).

	Constructs	Standardized Factor Loadings	Composite Reliability	Average Variance Extracted	Cronbach’s Alpha
1.	MSN	0.60–0.84	0.89	0.52	0.90
2.	DI	0.69–0.84	0.85	0.59	0.85
3.	Liking	0.78–0.93	0.91	0.77	0.90
4.	Habits	0.88–0.91	0.92	0.80	0.92
5.	Need and hunger	0.64–0.83	0.80	0.58	0.79
6.	Health	0.80–0.90	0.89	0.73	0.88
7.	Convenience	0.66–0.92	0.86	0.69	0.85
8.	Pleasure	0.78–0.90	0.87	0.69	0.86
9.	Traditional eating	0.58–0.74	0.70	0.44	0.70
10.	Natural concerns	0.82–0.87	0.89	0.73	0.89
11.	Sociability	0.67–0.90	0.86	0.67	0.84
12.	Price	0.85–0.88	0.90	0.76	0.90
13.	Visual appeal	0.77–0.86	0.87	0.69	0.87
14.	Weight control	0.55–0.88	0.80	0.58	0.71
15.	Affect regulations	0.86–0.93	0.92	0.79	0.91
16.	Social norms	0.69–0.85	0.84	0.64	0.83
17.	Social image	0.68–0.87	0.81	0.59	0.81
18.	NK	_	_	_	0.82
19.	Motivation (2d order CFA)				
	Liking	0.808			
	Habits	0.694			
	Need and hunger	0.809			
	Convenience	0.645			
	Pleasure	0.837			
	Traditional eating	0.841			
	Sociability	0.698			
	Price	0.700			
	Visual appeal	0.656			
	Affect regulations	0.441			
	Social norms	0.421			

**Table 3 nutrients-14-01547-t003:** Adjustments for measuring structural and reduced models (*n* = 807).

	Acceptable Fit Thresholds	Measuring Model	Structural Model	Reduced Model
X^2^ (df)	-	4005.685	4180.992	2541.846
		(1564)	(1566)	(967)
X^2^/df	<3.00–5.00	2.561	2.669	2.628
RMSEA	<0.05–0.08	0.044	0.049	0.045
CFI	>0.90	0.904	0.898	0.922
TLI	>0.90	0.899	0.892	0.916

**Table 4 nutrients-14-01547-t004:** Indirect, direct, and total standardized effects of determinants on intention to discard.

Dimensions			Indirect Effects	Direct Effects	Total Effects
Independent Variable	Mediator	Dependent variable			
NK	MSN	DI	0.023	0.038	0.061 *
EM	MSN	DI	0.061 ***	−0.099 ***	−0.038

Note: *,*** significant at levels 0.10 and 0.01, respectively.

## Data Availability

Restrictions apply to the availability of these data. Data was obtained from interviewees and are available from the corresponding author with the permission of the interviewees.

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
