# Peer review of "Effect of Warning Labels on Consumer Motivation and Intention to Avoid Consuming Processed Foods"

_nutrients, 2022, doi:10.3390/nu14081547_

Round 1

Reviewer 1 Report

Abstract. Perfect explained

Introduction is perfect: complete and very well structured

Reseach and methodology are well explained but results are shown in a complicated way. Is possible any graphic or more simple table?

It is a significance and original research but with a limit: Chilean context. For this reason, it is necessary explain why is important the results in Chilean context and other contexts.

Reviewer 2 Report

This paper focuses on an interesting topic on how warning labels are affecting consumer motivation and intention of avoiding processed food. However, the authors should be more cautious to build the conceptual model and put up hypothesis. Some of the detailed comments are listed below.

  1. The authors build up several (7) hypothesis based on previous literature, but some of them are not convincing. There is no clear explanation that the message sent by NWs can be a mediating factor between motivation and intention to discard (H4), or nutritional knowledge and intention to discard (H7).
  2. As the authors suggested, nutritional knowledge is mostly prior knowledge and varies among consumers, it would be more appropriate to investigate how NWs are affecting the intentions of consumers of different knowledge level or focus on how NWs can update the effects of nutritional knowledge on avoiding intention.
  3. Line 310-316, many subdimensions represent EM in the model, did you include them all in the empirical model or reduced the dimensions?
  4. Table 2, what is CN? Is it Nutritional knowledge??
